# A Systematic Preparation of Liposomes with Yerba Mate (*Ilex paraguariensis*) Extract

**DOI:** 10.3390/plants14091325

**Published:** 2025-04-28

**Authors:** Yasmine Miguel Serafini Micheletto, Brenda Vieira de Jesus, Gisele Louro Peres, Vânia Zanella Pinto

**Affiliations:** 1Engenharia de Alimentos, Universidade Federal da Fronteira Sul, Campus Laranjeiras do Sul, BR 158 km 405, Caixa Postal 106, Laranjeiras do Sul 89815-899, PR, Brazil; yasmine.micheletto@uffs.edu.br (Y.M.S.M.); brendavieiradjesus@gmail.com (B.V.d.J.); gisele.louro@uffs.edu.br (G.L.P.); 2Programa de Pós-Graduação em Ciência e Tecnologia de Alimentos (PPGCTAL), Universidade Federal da Fronteira Sul, Campus Laranjeiras do Sul, BR 158 km 405, Caixa Postal 106, Laranjeiras do Sul 89815-899, PR, Brazil

**Keywords:** Aquifoliaceae, *Ilex paraguariensis*, chlorogenic acids, phytosome, lipid vesicles

## Abstract

Lipid vesicles, liposomes and phytosomes have been gaining significant attention in various applications for phytochemical preservation. Furthermore, yerba mate (*Ilex paraguariensis*) contains a high content of bioactive compounds with functional properties; however, its liquid extract exhibits limited stability. For the first time, lipid vesicles containing yerba mate extract were produced and characterized. They were prepared using pure or purified phosphatidylcholine (PC) and n-hexane as a solvent via the reverse phase evaporation method. Their characterization was conducted using Fourier transform infrared spectroscopy (FTIR), UV–vis spectroscopy, Zeta potential (PZ), and dynamic light scattering (DLS). The decrease or absence of FTIR bands and UV–vis absorbance (325 nm) from the yerba mate extract suggests the successful dispersion of yerba mate extract in the liposome membrane, ensuring its encapsulation or complexation. Additionally, the size of lipid vesicles decreased from 625.1 nm to 440.5 nm (pure PC) and from 690.0 nm to 518.6 nm (purified PC) after the addition of yerba mate extract PZ values showed a slight change in all vesicles enhancing colloidal stability. This, combined with the reduction observed in DLS, suggests membrane reorganization, leading to the formation of unilamellar liposomes. Our observations indicate the possible formation of phytosomes, although additional studies are necessary to confirm this mechanism.

## 1. Introduction

Liposomes, or lipid vesicles, consist of a core containing an aqueous volume and one (or more) lipid bilayers, also known as membranes, formed by lipid molecules. These vesicles have been widely used as delivery systems due to their ability to encapsulate hydrophilic substances in the aqueous core and hydrophobic substances in the lipid bilayer [1,2]. Because of their lipid composition, liposomes are regarded as having reduced toxicity and are considered less toxic, non-immunogenic and biodegradable [3]. Consequently, they hold significant potential for food applications, particularly for the encapsulation, release, or dispersion of functional components including proteins, enzymes, antimicrobial agents, antioxidants, and flavorings and phytochemicals [4,5,6]. The use of liposomes in food has attracted increasing attention due to their physicochemical properties of kinetic stability and biocompatibility with food constituents. Furthermore, liposomes can enhance the flavor properties of various natural additives by encapsulating volatile or degradation-sensitive metabolites. Additionally, they can help protect food from spoilage and degradation [5,7].

Just as liposomes have the ability to encapsulate and deliver bioactive compounds in food, they offer significant potential for enhancing the effectiveness of natural ingredients [8], like those from yerba mate (*Ilex paraguariensis,* A. St.-Hil. (Aquifoliaceae)). Rich in phenolic acids, flavonoids, and methylxanthines, yerba mate provides various health benefits, including antioxidant, antimicrobial, and diuretic effects [9]. In particular, yerba mate has a high content of water-soluble chlorogenic acid isomers, rutin, and caffeine, which contribute to its antioxidant, anti-inflammatory, and hepatoprotective properties [9,10,11]. These bioactive compounds have been demonstrated to positively affect the management of conditions like obesity and diabetes [12]. However, their effectiveness can be compromised when exposed to light and heat or when dissolved in solutions, leading to degradation [13,14].

Given these challenges, encapsulating the bioactive compounds of yerba mate plays a key role in preserving their potency and ensuring their stability. For example, encapsulating yerba mate phenolic compounds using zein as a carrier through spray-drying or anti-solvent precipitation followed by freezer-drying has been shown to preserve over 95% of chlorogenic acids in yerba-mate co-product extracts [13,14]. The encapsulation approach provides to compounds some protection against degradation. It makes them more accessible for incorporation into food products, providing a valuable nutrient for nutraceutical uses [15], and to improve food shelf life [16].

Similarly, as carrier systems, liposomes have been demonstrated to improve the oral bioavailability of chlorogenic acid, a key phenolic compound in yerba mate, while enhancing its in vivo antioxidant activity [17]. The same study also revealed that liposomes facilitated the accumulation of chlorogenic acid predominantly in the liver of mice, suggesting hepatic targeting potential and contributing to its hepatoprotective effects. By utilizing lipid vesicle delivery systems, chlorogenic acids were better protected, stable for 6 months at 2–8 °C or 25 °C, and showed enhanced permeability and oral bioavailability [8]. Additionally, the phospholipid complex improved the protection of chlorogenic acid against UVA induced oxidative stress [18].

Phytosomes are also lipid vesicles used to improve the bioavailability and stability of phytochemicals. Its synthesis involves the formation of stable molecular complexes, mainly through hydrogen bonding, between phytochemicals and the phosphate head groups of phospholipids in the liposome membrane. These interactions offer a stable and controlled delivery system for water-soluble compounds like chlorogenic acids, further enhancing their therapeutic potential [8]. Due to these properties, these lipid-based systems are valuable tools for effectively incorporating crude extracts, including from yerba mate, in functional foods and nutraceuticals [19,20]. Food-grade phytosomes were developed using *Cyclopia subternata* crude extract, rich in C-glucosyl xanthones, benzophenones, and dihydrochalcones. The lipid vesicles protected the extract compounds for 6 months, when freeze-dried samples were stored at 25 °C and 40 °C at low relative humidity [19].

In this context, liposomes/phytosomes hold potential as encapsulation systems for preserving bioactive compounds, including those from yerba mate extract. Therefore, the aim was, for the first time, to prepare lipid vesicles containing yerba mate extract dispersed in the lipid bilayer and characterize them using techniques such as Fourier transform infrared spectroscopy (FTIR-ATR), UV–vis spectroscopy, dynamic light scattering (DLS) and Zeta potential (PZ).

## 2. Results and Discussion

### 2.1. FTIR and UV–Vis Characterization of Liposome for Solvent Selection

The FTIR spectra of liposomes prepared using pure phosphatidylcholine (PC) with chloroform (black line) or n-hexane (red line) as a solvent in the first step of the liposome preparation process revealed no significant differences (Figure 1A). The choice of solvents is based on their characteristics, such as solubility, polarity, and volatility. n-hexane, being food grade, also meets safety requirements for both food and pharmaceutical applications. Notably, the organic solvent is used only in the first step of the liposome preparation process to solubilize phospholipids. After this step, the solvent is evaporated using a rotary evaporator, leaving only liposomes suspended in water [1,21]. However, traces of solvent may remain. By using the n-hexane, we ensured the complete absence of chloroform residues.

In addition to conventional liposome formulations, phytosomes have been developed to enhance the encapsulation and delivery of bioactive compounds. Quercetin was complexed into phytosomes using olive oil, which were first dissolved in acetone and subsequently in n-hexane. This composition resulted in very stable vesicles, up to three months (4 °C and 25 °C), and high skin permeation of quercetin [22].

Alternatively, ethanol was also explored as a solvent in the preparation of lipid vesicles (ethosomes). This was achieved by solubilizing PC in ethanol followed by gradual addition of distilled water until reaching a final ethanol-to-water ratio of 70:30 (*v*/*v*). Caffeic acid-loaded ethosomes were prepared by active compound into the PC-ethanol solution before water addition. Transmission electron microscopy revealed a characteristic multilamellar structure. Furthermore, this delivery system was shown to be safe and non-irritating upon skin application [23]. These findings highlight the potential of using alternative solvents in the preparation of liposomes and phytosomes.

The absence of notable variations in FTIR spectra indicates that the chloroform or n-hexane solvent also did not affect the molecular structure of the liposomes (Figure 1A). The antisymmetric stretching vibration of the phosphate group (PO_2_⁻), typically observed in the frequency range of 1260–1220 cm⁻^1^ [24], provides insight into the molecular interactions within the liposome. The presence of a broad shape and weak peak in this frequency range (Figure 1A, blue arrow) may indicate a heterogeneous distribution of chemical environments for the phosphate group, likely due to hydration or interactions with other components in the system. In addition, intermolecular interactions between phosphate groups of different phospholipid molecules could also involve further interactions with water, residual solvents, or other lipid molecules.

On the other hand, the carbonyl stretching, observed in the frequency range of 1725 to 1740 cm⁻^1^ (C=O), which is characteristic of the fatty acid chains in phosphatidylcholine, did not show any shift (Figure 1A). Carbonyl stretching vibration is sensitive to the chemical environment and interactions between lipid chains, particularly with water molecules or other nonpolar molecules. Therefore, the band in this range can provide insights into the order of the lipid chains within the liposomal bilayer. Shifts in frequency or changes in the intensity of this band may indicate lipid–lipid interactions or phase transitions (e.g., between gel and liquid-crystalline phases). The shifting to lower frequencies could suggest a stronger interaction between the carbonyl group and water, potentially altering the lipid bilayer structure [24,25,26].

Additionally, the symmetric and antisymmetric stretching vibrations of the methylene groups in the acyl chains of the lipids occur at around 2850 cm⁻^1^ (CH_2_) and 2920 cm⁻^1^ (CH_2_), respectively. These vibrations are well known and are commonly used to characterize the conformational state of the lipid chains. They may overlap with the broadband range of 3000 to 3500 cm⁻^1^ [27], as seen in Figure 1A. This overlap suggests that a significant number of water molecules are associated with the lipid bilayer of the liposomes. Water molecules likely interact with the polar heads of PC, primarily the phosphate and choline groups, forming a network of hydrogen bonds. These interactions contribute to broadening the band associated with the O-H stretching of water molecules.

Interestingly, the antisymmetric stretching vibrations of choline, typically observed around 970 cm⁻^1^, were not detected in the FTIR spectra (Figure 1A). This absence may indicate that the choline headgroup is involved in significant intermolecular interactions, such as hydrogen bonding with water molecules or forming more complex structures with other components in the bilayer. These interactions could alter or reduce the intensity of the band, making it less visible or harder to detect in the spectrum. Another possibility is that the band is overlapped by other low-energy vibrations present in the spectrum. In liposomal systems, the organization and packing of the polar headgroups can vary, which directly influences the visibility of certain bands [27]. Furthermore, the absence of the choline band indicates that the polar headgroups are highly organized or participate in interactions that hinder their detection by FTIR.

Simultaneously, UV–vis spectra of liposome containing pure PC, prepared with chloroform or n-hexane as solvents, showed no absorption bands (Figure 1B). This behavior was expected, as the phospholipids in the liposomal bilayer do not absorb in the UV–vis range. Finally, using different solvents (chloroform and n-hexane) in the liposome preparation did not affect the chemical structure of the vesicles, as observed in Figure 1A,B. Therefore, n-hexane was chosen to prepare the liposome samples due to its food-grade status.

### 2.2. FTIR of Liposomes With or Without Yerba Mate Extract

FTIR analysis reveals some interaction between the yerba mate extract and the PC liposomes prepared with different volumes of extract (200 µL, 500 µL, and 1000 µL) (Figure 2). Upon incorporation of the extract into the liposomes, many of the characteristic peaks associated with the extract disappear, and the FTIR spectrum closely resembles that of the pure liposomes.

The numerous peaks correspond to functional groups of molecules present in the yerba mate extract, making their individual identification difficult [11]. Importantly, the absence of these bands in the liposome samples suggests that the yerba mate extract molecules are incorporated into the liposomal membrane, where they likely interact with the lipid bilayer rather than remaining free in solution. One possible explanation for this observation is that the polar groups in the yerba mate extract interact with the polar heads of the phospholipids through hydrogen bonding.

Purified PC derived from crude soy lecithin was used to prepare liposomes containing yerba mate extract. The purification process separates fatty acids, resulting in a mixture of phospholipids, predominantly phosphatidylcholine molecules with varying hydrocarbon tail lengths and degrees of unsaturation. The “pure” phosphatidylcholine used in this study (Across Organics) also consists of phosphatidylcholine molecules with different tail lengths and unsaturation concentrations.

The FTIR spectra of liposomes prepared with purified PC and different volumes of yerba mate extract (200 µL, 500 µL, and 1000 µL) further support the interaction between the extract and the liposomal membrane (Figure 3). As observed in our previous results with pure PC (Figure 2), the spectra of all liposome samples overlapped, and the characteristic bands of the yerba mate extract were absent in the liposome containing the extract. This suggests that the molecules from the yerba mate extract are incorporated into the lipid bilayer and are not free in solution, consistent with previous observations of extract encapsulation in liposomes. Additionally, there were no differences in the FTIR spectra of liposomes prepared with pure or purified PC, reinforcing the idea that the purification process does not alter the lipid structure. These results confirm that purified PC, which is cost-effective, can be used for liposome preparation and for studying interactions with target molecules.

Nanophytosomes containing chlorogenic acid, the primary phenolic acids in yerba mate, were produced by solvent evaporation technique using two phospholipids, Phospholipon^®^ 90H or Lipoid^®^ S100. The ^1^H-NMR results show the disappearance of distinct chlorogenic acid peaks from chlorogenic acid-loaded nanophytosomes. These changes suggest the formation of intermolecular interactions between the chlorogenic acids’ OH groups and the polar groups of both the phospholipids, indicating hydrogen bonding. The phenolic OH groups were bound to the choline N [CH_3_] and phosphate (P=O) moieties of both the phospholipids. The disappearance and shifts in the characteristic FTIR absorption bands corresponding to these functional groups further confirm the presence of hydrogen bonding, van der Waals forces and ion dipole forces between chlorogenic acid and the phospholipids, leading to the formation of nanophytosomes [8].

### 2.3. UV–Vis of Liposomes With or Without Yerba Mate Extrac

The UV–vis absorption band at 325 nm, characteristic of chlorogenic acids, is prominent in the yerba mate extract (Figure 4, black line), indicating a high concentration of free chlorogenic acids in the solution. These compounds, abundant in yerba mate, are responsible for the observed absorption at this wavelength [10,11]. However, this absorption was not observed in the liposomes without (Figure 4, blue line) or containing 200 µL of yerba mate extract (Figure 4, pink line), suggesting that the chlorogenic acids from the extract were incorporated into the lipid bilayer and are interacting with the phospholipids of the liposomal membrane. This absence may also indicate that the interaction of chlorogenic acids with the lipids alters their electronic environment, thus suppressing the characteristic absorption at 325 nm. This effect is commonly observed when phenolic compounds are successfully encapsulated.

In contrast, liposomes containing 500 µL (Figure 4, green line) and 1000 µL of yerba mate extract (Figure 4, red line) still exhibited the characteristic absorption band of chlorogenic acids, though with reduced intensity compared with the free extract. This suggests that the encapsulation capacity of the lipid bilayer may have been exceeded, resulting in unincorporated compounds that are free in solution or weakly associated with the liposome surface. Such unencapsulated or surface-bound compounds can still influence the behavior and properties of liposomes. Molecules adsorbed onto the liposome surface may affect colloidal stability, modulate interactions with cell membranes, or alter bioavailability. In general, high encapsulation efficiency is desirable, and additional steps are often taken to remove unencapsulated drugs and organic solvents to ensure product safety [28]. However, in some cases, the presence of free compounds may add functional versatility to food or cosmetic formulations, working synergistically with the encapsulated fraction to produce both immediate and sustained effects.

UV–vis also confirmed chlorogenic acids loading into an apoferritin complex with a sodium alginate-apoferritin co-encapsulation system. A shift in the absorption peak at 280 nm indicated an interaction between chlorogenic acids and amino acid residues in the inner ferritin cage [29]. Liposomes were used to encapsulate active pharmaceutical ingredients, and their encapsulation efficiency was confirmed by UV–vis spectroscopy. The absorption spectra of substances at different liposome processing steps revealed the vesicles shielding effect, with no phospholipids remaining in the hydrophobic phase [30].

Furthermore, the liposomes efficiently encapsulated the yerba mate extract. However, when larger extract volumes were added, the compounds continued interacting with the lipid bilayer but were not fully dispersed within the liposomal membrane (Figure 4 and Figure 5).

To further assess the influence of purified PC on the encapsulation process, liposomes were evaluated for varying volumes of yerba mate extract (Figure 5). The UV–vis spectra reveal behavior similar to those produced using pure PC (Figure 4). Lower volumes of year mate extract were successfully encapsulated, while higher volumes were not completely dispersed within the lipid bilayer membrane. These results indicate that the purified PC is as efficient as pure PC in loading yerba mate extract. This in turn led us to hypothesize that the interactions with molecules from the yerba mate extract and lipid bilayer membrane appear to be limited by the availability of the choline headgroups [2]. Additionally, these headgroups are involved in significant intermolecular interactions, such as hydrogen bonding with polar molecules (e.g., water, chlorogenic acids, rutin) or in forming more complex structures with other components in the bilayer. Furthermore, the yerba mate extract could contribute to the self-assembly of the liposomes into unilamellar vesicles, a process enhanced by sonication during preparation, which results in smaller liposomes. In this way, the low volume of yerba mate extract (200 µL) ensures that the liposomes load the whole extract content.

### 2.4. Potential Zeta (PZ) and Dynamic Light Scattering (DLS)

The PZ of liposomes prepared with pure PC was measured at −46.8 mV, whereas for those prepared from purified PC it was −67.5 mV (Table 1). In general, colloidal systems with PZ outside the ±30 mV range typically demonstrate enhanced electrostatic stability due to strong interparticle repulsive forces [8,21].

In our study, the observed negative PZ confirms that electrostatic repulsion between liposomal vesicles plays an important role in maintaining colloidal stability. These repulsive forces prevent particle aggregation or fusion but also ensure stability during diffusion and storage. This highly negative value may be related to the presence of phosphatidic acid, an anionic phospholipid, abundant in the composition of crude soybean lecithin [21].

Unlike PC, which is a zwitterionic phospholipid, phosphatidic acid contributes a single negative charge, significantly amplifying the overall negative surface charge of the liposomes [31]. This compositional variation illustrates how different lecithin sources can affect the liposomes’ surface charge, with potential implications for biological interactions [2]. This characteristic is relevant because the presence of additional negative charges can enhance the colloidal stability of the liposomes, preventing aggregation and promoting uniform dispersion in aqueous suspensions. Furthermore, the negative charge associated with phosphatidic acid may influence the interactions of the liposomes with other molecules, allowing for better control over the release of encapsulated substances [31].

Liposomes prepared with pure PC or purified PC with 200 µL of yerba mate extract showed high negative PZ (Table 1). The PC source did not influence the PZ when the yerba mate extract was added to the liposomes.

A study of caffeic acid inclusion in liposomes resulted in a less negative PZ, with an absolute reduction of 18 mV. This shift toward a nearly neutral PZ suggests a balanced distribution of positive and negative charges on the vesicle surface [23]. Similarly, nanophytosomes were used to improve the stability of *Cyclopia subternata* crude extract. The PZ varied from −38.9 to −46.4 mV across 20 treatments. The high absolute PZ suggests a strong electrostatic repulsion between lipid vesicles, indicating a theoretically high colloidal stability of the solution [19].

Previous investigation into other liposomes with polyphenol revealed that green tea waste catechins, when added in single- or double-layered liposomes, formed stable colloidal suspensions for 28 days at 4 °C. The PZ ranged from −8.31 to −7.89 mV for single-layer liposomes and −8.98 to −8.11 mV for double layer when incorporated into kiwifruit juice. The absence of changes in surface charge suggested that catechins from green tea waste were likely included at low concentrations, which may have limited their impact on PZ [32]. These findings suggest that the magnitude of phenolic compound–PC interactions and their effects on surface charge are strongly influenced by the concentration and chemical nature of the phenolic compounds more than the PC source.

Loaded-liposome characteristic changes may be related to the reorganization of the liposomal membrane in the presence of phenolic compounds from the yerba mate extract within the lipid bilayer. These changes in PZ may be associated with the reorganization of the liposomal membrane, as the phenolic compounds from yerba mate extract induce a different arrangement of the lipid bilayer, altering the charge distribution. Nevertheless, the negative value still indicates colloidal stability in suspension [8,21,33].

Although PZ values confirm electrostatic stabilization, they alone do not comprehensively define the colloidal properties of the liposomal system. Liposomes prepared with pure PC or purified PC exhibited a hydrodynamic diameter > 625 nm and a PDI > 0.434, while, when containing yerba mate extract (200 uL), they showed a decreased size < 518 nm and a PDI > 0.589 (Table 1). The reduced size and PDI from pure PC liposomes reflect moderate heterogeneity and are consistent with the characteristics of the reverse-phase evaporation method [21]. The purified PC forms relatively large liposomes; however, this can be beneficial, as larger liposomes often have a greater capacity to encapsulate larger or multiple molecules, depending on the application.

DLS analysis also revealed two populations for all liposomes with a predominance between 500–750 nm (Figure 6A–D), indicating the coexistence of unilamellar (smaller) and multilamellar (larger) vesicles, as reported for analogous systems [19,20]. Multilamellar liposomes have multiple lipid bilayers (several lamellae) and, as such, tend to have larger diameters. Therefore, the concentration of phosphatidylcholine, as well as the sonication assisting, temperature, and evaporation rate, may strongly influence the distinct population sizes.

Notably, yerba mate extract incorporation reduced liposome diameter by ~30% (to 440.5 nm), suggesting interactions between phenolic compounds and phospholipid headgroups. This size reduction (29.5%) exceeds the 20–25% reported for olive leaf phenolic compounds [34]. These liposomal membrane reorganizations can be due to the presence of phenolic compounds abundant in yerba mate extract, particularly the most abundant chlorogenic acids, rutin, and caffeine [10].

This size versatility enables diverse applications across multiple sectors. In pharmaceuticals, small-size vesicles serve as stable carriers, making them ideal for drug delivery. In cosmetics, larger liposomes may improve skin retention and improve topical efficacy. For nutraceuticals, intermediate-sized vesicles are suited for bioactive compounds. Similarly, food applications benefit from intermediate-sized liposomes, which provide stable colloidal dispersions for beverages and contribute to desirable viscosity in functional food products [3].

Given the reduction in hydrodynamic diameter, the colloidal stability, and the presence of abundant phenolic compounds in yerba mate extract, we hypothesize that a phytosome-like structure may have formed. This is based on the potential for hydrogen bonding interactions between polyphenols (e.g., chlorogenic acids, rutin) and the phosphatidylcholine headgroup, which could stabilize unilamellar vesicles and prevent multilamellar assembly. Furthermore, phytosomes are smaller in size and have only one lipid bilayer [18]. These interactions between phenolic compounds and PC headgroup also support the membrane dispersion, resulting in an efficient encapsulation and stability of the bioactive compounds, as confirmed by FTIR and UV–vis results. Although our data suggest this structural organization, further studies involving spectroscopic or microscopic techniques are necessary to confirm the formation of phytosomes. Moreover, phytosomes not only facilitate the controlled delivery of bioactive molecules but also enhance the solubility and bioavailability of these compounds, which can be beneficial in several applications, especially in beverages and food products, cosmetics, drugs and for agriculture uses.

## 3. Materials and Methods

### 3.1. Materials, Yerba Mate Extract, and Phosphatidylcholine (PC) Preparation

The yerba mate was purchased from a local Laranjeiras do Sul PR market. The solvents used were 95% ethanol, n-hexane PA, and chloroform PA (Merck KGaA, Darmstadt, Germany). The extract was prepared by mixing 0.058 g of yerba mate with 15 mL of 75% ethanol (*v*/*v*) and heating the mixture in a water bath at 80 °C for 3 h, stirring at 150 rpm. Afterward, the mixture was filtered to separate the insoluble solids [10], and the soluble fraction was used for liposome preparation. The chemical composition of the yerba mate extract was similar to our previous report using 75% ethanol as solvent [10].

Cholesterol (≥99%) was obtained from Sigma Aldrich (St. Louis, MO, USA). Pure phosphatidylcholine (PC) was sourced from Across Organics (Geel, Belgium), and used without prior treatment.

The soy lecithin used to obtain purified PC was provided by Gebana Brasil (Capanema, PR, Brazil). The purification of the soy lecithin was carried out as described by Mertins et al. [21]. First, 10 g of soy lecithin was dissolved in 50 mL of ethyl acetate, and 2 mL of distilled water was added under stirring, forming two phases. The gel phase was then separated from the liquid phase, dispersed in 30 mL of acetone, forming aggregates, and ground using a glass stick. The granular phase was removed through filtration, and a new portion of 30 mL of acetone was added, repeating the grinding process. The precipitate was then vacuum-filtered and stored in a desiccator. This separation process eliminates free fatty acids in crude soy lecithin. The molecular composition of the purified PC following this method consists of approximately 75% di stearoyl phosphatidylcholine (DSPC), 12% dioleoyl phosphatidylcholine (DOPC), and 8% dipalmitoyl phosphatidylcholine (DPPC) [21].

### 3.2. Lipossome Preparation

Liposomes were prepared using a reverse-phase evaporation method. First, pure PC or purified PC (0.04 g) and cholesterol (0.02 g) were dissolved in 10 mL of an organic solvent (chloroform or n-hexane). Next, 200 µL, 500 µL, or 1000 µL of yerba mate extract were added. To facilitate solubilization, a few milliliters of ultrapure water were added to the suspension, which was then sonicated for 2 min to promote the formation of reverse micelles. Then, the organic solvent was evaporated using a rotary evaporator, yielding a lipid film, known as organogel. Finally, ultrapure water was added to the film under constant stirring to form a liposome suspension at a concentration of 15 mg/mL [21].

### 3.3. Fourier Transform Infrared Spectroscopy (FTIR)

The FTIR spectra were collected using an ATR-FTIR spectrometer (IRTracer-100, Shimadzu Corp., Kyoto, Japan) for liposomes loaded or not with yerba mate extract. The measurements were made in triplicate at 400–4000 cm^−1^ with a resolution of 4 cm^−1^ and with 100 scans at room temperature.

### 3.4. UV–Vis Spectroscopy

UV–vis spectra were obtained using a UV–vis spectrophotometer (Thermo Scientific, Multiskan GO, Waltham, MA, USA) from liposomes, with or without yerba mate extract. The measurements were performed in triplicate in a quartz cuvette with 1 cm of optical path between 200–600 nm at room temperature.

### 3.5. Potential Zeta (PZ) and Dynamic Light Scattering (DLS)

The PZ analysis of the liposomes was performed using the Zetasizer Nano series ZS90 (Malvern, UK) with a polystyrene cuvette (DTS0012) supplied with a 5 mW helium neon laser, with a wavelength output of 633 nm. The measurements were conducted in triplicate at room temperature and each individual measurement was an average of 12 scans. The Zeta potential and SD were calculated automatically using the electrophoretic mobility of the lipid vesicles in solution using the Smoluchowski equation. The hydro-dynamic diameter and polydispersity index (PDI) of the liposomes were also determined using dynamic light scattering (DLS) under the same conditions using capillary cells for Zeta potential (DTS1070).

## 4. Conclusions

In this study, liposomes, with or without yerba mate extract, were prepared using the reverse phase evaporation technique, employing either pure or purified PC and two different solvents during the initial preparation step. No differences in the FTIR and UV–vis spectra were observed between the two forms of PC or the solvents used. The FTIR-ATR and UV–vis analyses suggest that the compounds in the yerba mate extract interact with the liposomal lipid bilayer. The decrease in size of liposomes containing 200 µL of yerba mate extract points to a reorganization of the liposomal membrane, likely due to phenolic compounds in the extract.

Based on these findings, we hypothesize that the interaction of phenolic compounds from yerba mate extract, such as chlorogenic acids and rutin, with the polar heads of PC leads to the formation of phytosome vesicles. These interactions, mediated by hydrogen bonds, may inhibit the formation of a second lipid bilayer, resulting in unilamellar liposomes with a smaller hydrodynamic diameter. Additionally, the negative Zeta potential characteristic of these systems indicates greater colloidal stability, preventing particle aggregation. The phenolic compounds may further enhance this stability by reinforcing the negative surface charge and promoting homogeneous dispersion of the liposomes.

Based on our observations, we propose that phytosome structures may have formed. The reduced vesicle size and enhanced colloidal stability observed and PZ values support the hypothesis that phytosome formation occurs when liposomes are prepared with yerba mate extract. However, further studies are required to confirm this mechanism. Although the preliminary results are promising, additional studies are required to evaluate the actual potential of these lipid-based systems for the controlled delivery of bioactive compounds from yerba mate extract. Future studies would also benefit from accessing the stability of yerba mate extract encapsulated by liposomes under varying pH, environment conditions and temperatures.

## Figures and Tables

**Figure 1 plants-14-01325-f001:**
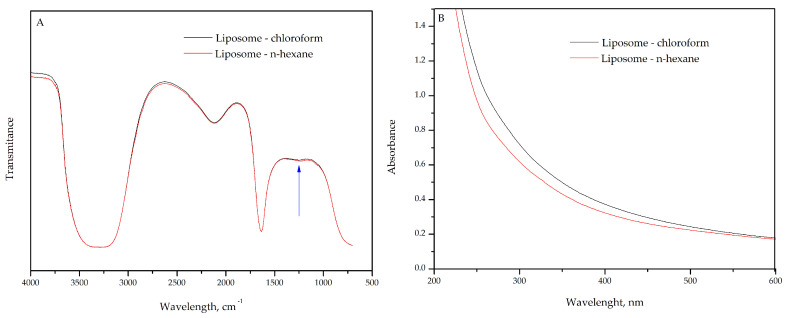
(**A**) FTIR-ATR spectrum and (**B**) UV–vis spectra of liposomes prepared with pure phosphatidylcholine (PC) and chloroform, and liposomes prepared with pure PC and n-hexane.

**Figure 2 plants-14-01325-f002:**
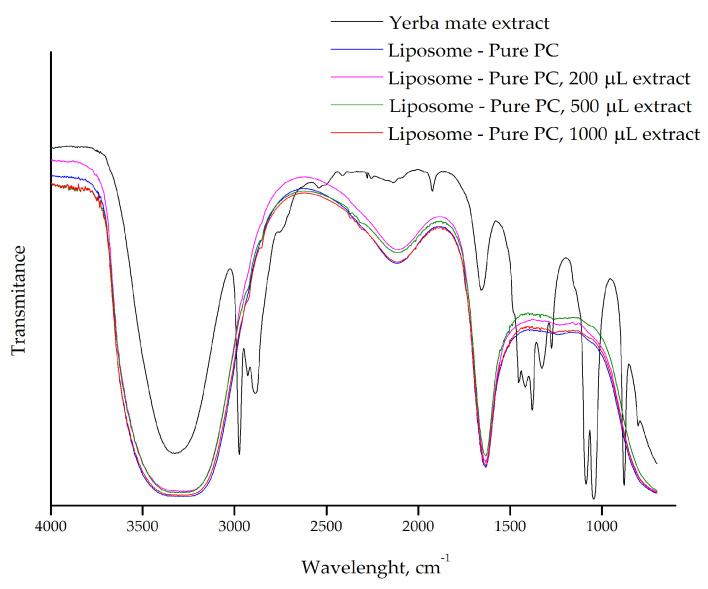
FTIR-ATR spectrum of yerba mate extract and the pure PC liposomes prepared with different volumes of extract (200 µL, 500 µL, and 1000 µL).

**Figure 3 plants-14-01325-f003:**
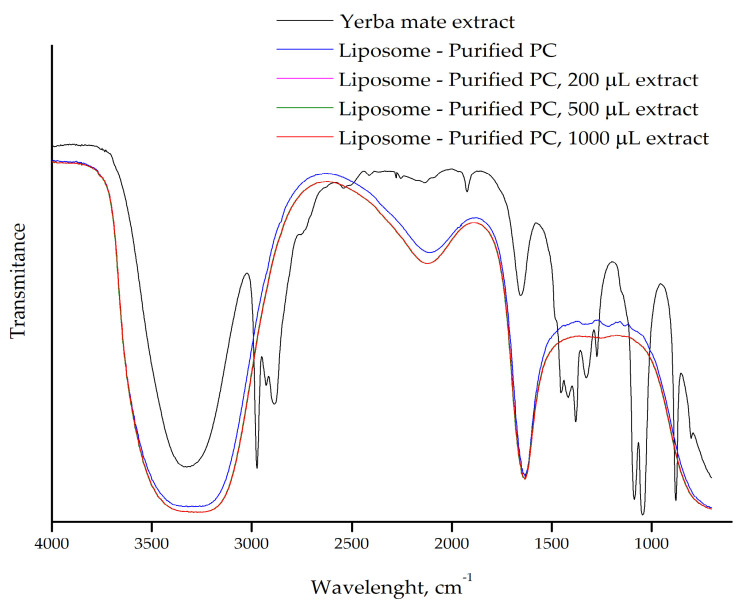
FTIR-ATR spectrum of yerba mate extract and the purified PC liposomes prepared with different volumes of extract (200 µL, 500 µL, and 1000 µL).

**Figure 4 plants-14-01325-f004:**
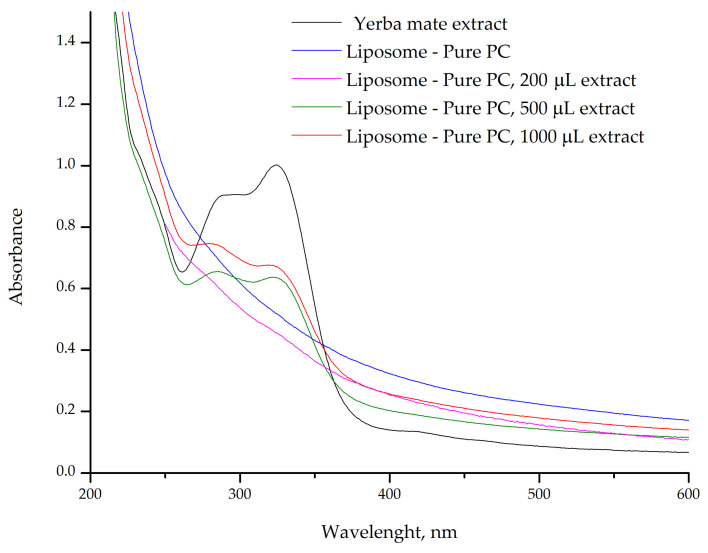
UV–vis spectrum of yerba mate extract and the pure PC liposomes prepared with different volumes of extract (200 µL, 500 µL, and 1000 µL).

**Figure 5 plants-14-01325-f005:**
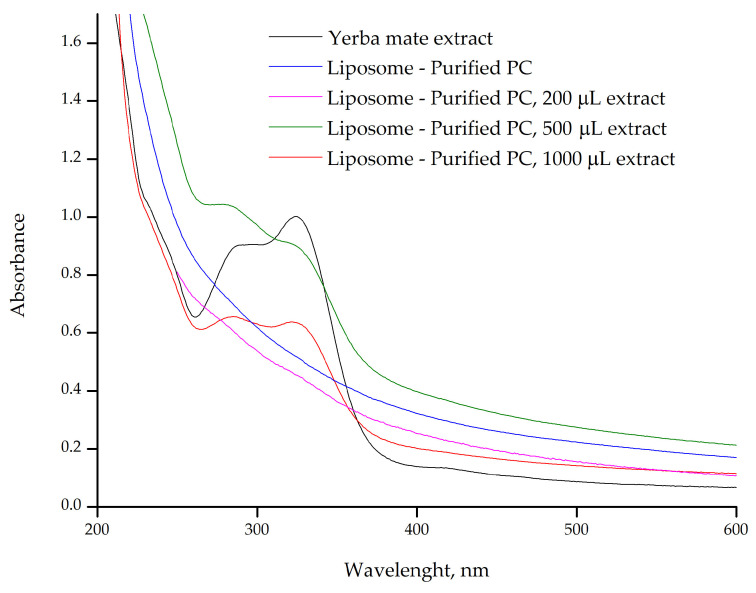
UV–vis spectrum of yerba mate extract and the purified PC liposomes prepared with different volumes of extract (200 µL, 500 µL, and 1000 µL).

**Figure 6 plants-14-01325-f006:**
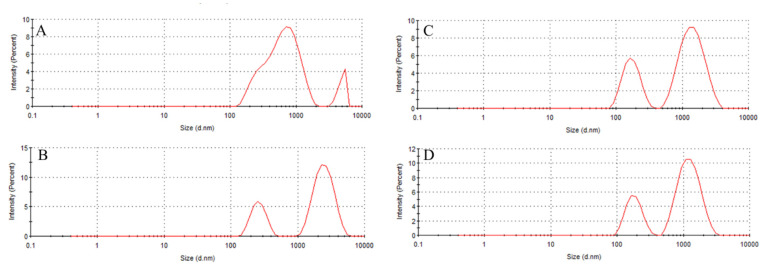
Light scattering intensity as a function of size distribution for the liposome prepared from (**A**) pure PC, (**B**) purified PC, (**C**) pure PC with yerba mate extract, and (**D**) purified PC with yerba mate extract.

**Table 1 plants-14-01325-t001:** Hydrodynamic diameter, polydispersity index (PDI), and Zeta potential (PZ) of liposomes formulated with pure PC or purified PC and yerba mate extract (200 uL).

Samples	PZ (mV)	Diameter (nm) ^1^	PDI
Liposome pure PC	−46.8 ± 1.8	625.1 ± 15.1	0.434
Liposome purified PC	−67.5 ± 0.9	690.0 ± 35.4	0.657
Liposome pure PC, yerba mate extract	−51.9 ± 3.2	440.5 ± 22.3	0.660
Liposome purified PC, yerba mate extract	−47.5 ± 0.5	518.6 ± 16.0	0.589

^1^ Hydrodynamic diameter.

## Data Availability

The original contributions presented in this study are included in the article. Further inquiries can be directed at the corresponding author.

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
