# Peer review of "A Systematic Preparation of Liposomes with Yerba Mate (Ilex paraguariensis) Extract"

_plants, 2025, doi:10.3390/plants14091325_

Round 1
Reviewer 1 Report
Comments and Suggestions for Authors
The authors study a practically important issue – the production of liposomal preparations of mixtures of biologically active substances from Ilex paraguariensis leaves. The topic of the article is of great importance for the food industry, pharmacology, and dietetics.
The article is written in clear English, the authors' thoughts are consistent (except for the Results section).
The article is poorly structured: the Results section contains a mixture of results with small Discussion elements and references to literature. Results should not contain methods, explanations, comparisons, or references to literature. There is no separate Discussion section. It is necessary to write a Discussion and divide it into 3-5 subsections with subheadings.
Technical comments.
- The words (Ilex paraguariensis) should be added to the title of the article.
- Why is the third author's last name written in capital letters?
- Line 18: the hyphen should be removed.
- The abstract is uninformative. Write the abstract so that when reading it in the Scopus and Web of Science databases, readers will want to read the entire article. The text should be more informative and more interesting. Add substances contained in Ilex paraguariensis to the abstract text.
- Add the word Aquifoliaceae to the keywords and to the abstract text.
- Line 43: the first mention should be accompanied by the author's surname and the name of the family in brackets - A.St.-Hil. (Aquifoliaceae).
- The Results and Materials and Methods sections should be swapped.
- Line 309: the GPS coordinates of the place where the plant was grown should be clearly indicated, since the chemical composition of the plant changes depending on the latitude.
- After each reagent that the authors used in the experiment, the % of pure substance in the reagent should be indicated.
- Line 342: 400–4000 should be written.
- Line 44-52: very superficial analysis of chemical composition. It would be better to add a table with names, structural formulas of substances, CAS number (e.g. 58-08-2), content of substances in dry leaves (%), physiological activity of substances and sources of literature.
- Figures 2-5 and others: a space is needed between the number and the unit of measurement in the legend, microliters should be written either in full or as µL.
- Table 1: statistical processing of data is missing. It is necessary to apply the method of multiple comparison of samples, for example the Tukey test.
- In the figures on the abscissa and ordinate axes, as well as in the table header, units of measurement should be written after the comma, not in brackets.
- Figures 2-5 should be made 50-60% wider. The height of the figures should not be increased.
- Figure 6 should be made 20% wider and 40% higher. All fonts in all figures should be approximately equal to the size of the letters in the text of the article.
- Line 354: the subsection "Statistical data processing" is missing.
- Literature with a large number of errors. Journal titles should be abbreviated in accordance with the international standard ISO 4. All words in article titles do not need to be capitalized.
Author Response
Reviewer #1
The authors study a practically important issue – the production of liposomal preparations of mixtures of biologically active substances from Ilex paraguariensis leaves. The topic of the article is of great importance for the food industry, pharmacology, and dietetics.
The article is written in clear English, the authors' thoughts are consistent (except for the Results section).
Re: We appreciate your time and consideration reviewing our manuscript. We have made all suggested corrections and highlighted them in red throughout the manuscript.
The article is poorly structured: the Results section contains a mixture of results with small Discussion elements and references to literature. Results should not contain methods, explanations, comparisons, or references to literature. There is no separate Discussion section. It is necessary to write a Discussion and divide it into 3-5 subsections with subheadings.
Re: Thanks for your comments. The section should be named as Results and Discussion (both together). We adjusted the section title, and the discussion was improved according to comments from all reviewers.
Technical comments.
- The words (Ilex paraguariensis) should be added to the title of the article.
Re: The title was improved as suggested.
- Why is the third author's last name written in capital letters?
Re: The author’s name was adjusted.
- Line 18: the hyphen should be removed.
Re: the typo problem was corrected.
- The abstract is uninformative. Write the abstract so that when reading it in the Scopus and Web of Science databases, readers will want to read the entire article. The text should be more informative and more interesting. Add substances contained in Ilex paraguariensis to the abstract text.
Re: The abstract has been rewritten to include key results and bioactive substances from Ilex paraguariensis, making it more informative..
- Add the word Aquifoliaceae to the keywords and to the abstract text.
Re: the word ‘Aquifoliaceae’ was added at keywords and at introduction section, to avoid extra words at abstract.
- Line 43: the first mention should be accompanied by the author's surname and the name of the family in brackets - A.St.-Hil. (Aquifoliaceae).
Re: The information was added as suggested.
- The Results and Materials and Methods sections should be swapped.
Re: the order of the information was set using the journal template and Methods section was the last one in the template.
- Line 309: the GPS coordinates of the place where the plant was grown should be clearly indicated, since the chemical composition of the plant changes depending on the latitude.
Re: The GPS coordinates from this plant material is not possible to achieve, because we used samples from an industry located in Laranjeiras do Sul, PR, Brazil, which process leaves cultivated in other cities and towns. However, yerba mate leaves are very well studied, and the extracts production is one of our main research lines. So, we include the reference of the previous extract production and characterization (Dos Santos et al., 2023).
- After each reagent that the authors used in the experiment, the % of pure substance in the reagent should be indicated.
Re: The information about the reagents were included in the Methods section.
- Line 342: 400–4000 should be written.
Re: Thanks for bring this typo problem. It was corrected.
- Line 44-52: very superficial analysis of chemical composition. It would be better to add a table with names, structural formulas of substances, CAS number (e.g. 58-08-2), content of substances in dry leaves (%), physiological activity of substances and sources of literature.
Re: The yerba mate compounds are very well studied from different research groups. Unfortunately, we did not have an chromatography available at the moment of the experiment to report an in deep chemical composition of the extracts, our equipment is not working. However, we use a previous stablished method from our research group to produce the extracts (Dos Santos et al., 2023), using less leaves weight to have a diluted extract. The focus was to study and provide an encapsulation method. In our future research we are exploring in deep extract stability and realising from phytosomes, and chemical composition will definitely be included and widely studied.
- Figures 2-5 and others: a space is needed between the number and the unit of measurement in the legend, microliters should be written either in full or as µL.
Re: The figures were adjusted as suggested.
- Table 1: statistical processing of data is missing. It is necessary to apply the method of multiple comparison of samples, for example the Tukey test.
Re: Re: Thank you for your comment regarding the replicates and statistical analysis in our DLS and zeta potential measurements. As noted in the Methods section, all analyses were performed following the manufacturer’s recommendations guidelines for nanomaterial characterization. The Zeta Sizer Nano Series (Malvern Instruments, UK) automatically conducts each measurement as an average of 12 consecutive scans to ensure reproducibility. For additional results reliability, we performed three independent replicates per sample (triplicate), with the final results reported as the mean ± standard deviation (SD) of these replicates.
Given the nature of our data in Table 1—where the measurements represent intrinsic physicochemical properties (e.g., size, PDI, zeta potential) rather than comparative experimental treatments—standard statistical tests (e.g., ANOVA or t-tests) were not applicable. This approach aligns with similar studies characterizing liposomes or nanoparticles, where triplicate measurements with mean ± SD are widely accepted to demonstrate trends and reproducibility (see references below). We appreciate the opportunity to clarify this point.
https://doi.org/10.3390/nano11010171
https://doi.org/10.1007/s00216-017-0527-z
https://doi.org/10.3390/molecules25235655
https://doi.org/10.1016/j.heliyon.2019.e02372
- In the figures on the abscissa and ordinate axes, as well as in the table header, units of measurement should be written after the comma, not in brackets.
Re: The figures were adjusted as suggested. However, table 1 wa kept units in brackets to avoid misunderstanding.
- Figures 2-5 should be made 50-60% wider. The height of the figures should not be increased.
Re: We did our best to wide while maintaining their original height to preserve the proportions, as requested.
- Figure 6 should be made 20% wider and 40% higher. All fonts in all figures should be approximately equal to the size of the letters in the text of the article.
Re: This figure is original from the Zeta sizer equipment. Unfortunately, this analysis was done in a different place, and we do not have access of the software to edit the image. However, it is possible to see some other publication with the same images strucutre from this kind of equipment. Furthermore, we made it wider and higher as much as possible to preserve proportions.
https://doi.org/10.3390/pharmaceutics13030390
https://www.thno.org/v06p0177.htm
https://doi.org/10.22088/cjim.13.1.90
https://doi.org/10.3389/fgstr.2024.1387343
- Line 354: the subsection "Statistical data processing" is missing.
Re: Thank you for your comment. The experiments were conducted under controlled and standardized conditions to minimize variability. Analyses of FTIR, and UV-vis focused on qualitative trends and structural characterization rather than quantitative variation. Statistical tests like ANOVA or t-tests are not applicable considering the nature of our data from these characterization techniques. However, Zeta potential equipment conducts 12 scans each replicate and delivery the mean of the 3 replicates ± SD. The diameter size also was analysed in triplicates and the SD was reported.
- Literature with a large number of errors. Journal titles should be abbreviated in accordance with the international standard ISO 4. All words in article titles do not need to be capitalized.
Re: The authors used Zotero reference manager set as MPDI Plants standard. However, all references were double checked to make sure they are properly standardized.
Reference used in this response:
Dos Santos, D. F., Alves, V., Costa, E., Martins, A., Vieira, A. F. F., Dos Santos, G. H. F., Francisco, C. T. D. P., & Pinto, V. Z. (2023). Yerba Mate (Ilex paraguariensis) Processing and Extraction: Retention of Bioactive Compounds. Plant Foods for Human Nutrition, 78(3), 526–532. https://doi.org/10.1007/s11130-023-01082-6
Reviewer 2 Report
Comments and Suggestions for Authors
This manuscript presents a valuable and structured study on the development of liposomes incorporating Ilex paraguariensis (yerba mate) extract using the reverse phase evaporation method. The work addresses a relevant topic within plant-based bioactives, with implications for food science, functional delivery systems, and nutraceutical research. The research is overall well-executed and documented. However, some revisions are needed, particularly in how certain hypotheses are presented and in the refinement of language and scientific claims.
The introduction clearly defines the research context and supports the scientific rationale. The authors provide a good overview of the biological activity of yerba mate, particularly its antioxidant potential, and briefly justify the interest in encapsulating phenolic compounds. The transition to liposomal technology is appropriate, although the link between phytosomes and conventional liposomes could be more clearly explained, especially since the term "phytosome" reappears in the conclusions without being conceptually developed earlier.
Additionally, the introduction would benefit from:
- A clearer statement of the knowledge gap, particularly what previous works have or have not achieved in liposome-based encapsulation of yerba mate.
- Inclusion of more recent references (2021–2023) to better frame the novelty of the work.
Moreover, the methodology section is well organized and detailed. The reverse phase evaporation technique is correctly applied, and the characterization steps (UV-Vis, FTIR, DLS, zeta potential, and encapsulation efficiency) are standard and appropriate.
Some improvements are suggested:
- In the preparation of extracts, it would be helpful to specify the standardization or characterization of the extract (e.g., total phenolic content or HPLC profile), to better relate it to its encapsulation performance.
- When describing DLS and zeta potential analysis, please include how many replicates were performed and whether statistical differences were evaluated.
- Clarify whether any controls or blank liposomes were tested in parallel and how they were used in interpreting results.
Also, the results are clearly presented and illustrated with appropriate figures. The size distribution, encapsulation efficiency, and zeta potential data are coherent and support the formulation's stability and encapsulation success.
However, some points in the discussion require clarification:
- The authors observe that the presence of yerba mate extract results in smaller vesicle size and increased negative surface charge, and they hypothesize an interaction between phenolics and the phospholipid headgroups that may inhibit multilamellar structure formation.
- While this is an interesting and plausible idea, there is no experimental data in the manuscript (e.g., microscopy, molecular modeling, or comparative encapsulation using purified compounds) to support this conclusion.
- Therefore, this hypothesis should be clearly stated as a theoretical interpretation or further justified with literature support or a sentence indicating the need for future confirmation.
- The term “phytosome” is introduced in the conclusion but not previously explained or defined in the introduction or methodology. If the authors suggest that this system behaves similarly to phytosomes due to interactions between phenolics and lipids, this should be clarified earlier in the manuscript and be conceptually separated from standard liposomes.
- Consider discussing potential applications or benefits of these findings in real-world systems (e.g., food, nutraceutical, or cosmetic delivery) to increase the impact of the work.
Furthermore, the conclusion summarizes the findings well but as mentioned above, includes a speculative mechanism (lines 364–367) that is not directly supported by the results. A clearer statement like “Based on our observations, we propose that such interactions may occur, although further studies are required to confirm this mechanism” would be more appropriate.
Comments on the Quality of English Language
The manuscript is understandable overall but would benefit from light editing for fluency and grammar. Some sentences are overly long or awkwardly structured. I recommend reviewing the manuscript with a native English speaker/colleague.
Author Response
Reviewer # 2
This manuscript presents a valuable and structured study on the development of liposomes incorporating Ilex paraguariensis (yerba mate) extract using the reverse phase evaporation method. The work addresses a relevant topic within plant-based bioactives, with implications for food science, functional delivery systems, and nutraceutical research. The research is overall well-executed and documented. However, some revisions are needed, particularly in how certain hypotheses are presented and in the refinement of language and scientific claims.
Re: We appreciate your time and comments. We reviewed carefully the manuscript and include as much detail as possible, as well new information suggested by all reviewers. Please see all changes highlighted in red throughout the manuscript.
The introduction clearly defines the research context and supports the scientific rationale. The authors provide a good overview of the biological activity of yerba mate, particularly its antioxidant potential, and briefly justify the interest in encapsulating phenolic compounds. The transition to liposomal technology is appropriate, although the link between phytosomes and conventional liposomes could be more clearly explained, especially since the term "phytosome" reappears in the conclusions without being conceptually developed earlier.
Re: Thank you for your observation. We included the explanation of phytosome concept, cited some research about it. Please see lines 72-83 at introduction section.
Additionally, the introduction would benefit from:
- A clearer statement of the knowledge gap, particularly what previous works have or have not achieved in liposome-based encapsulation of yerba mate.
- Inclusion of more recent references (2021–2023) to better frame the novelty of the work.
Re: We included some new researchers about phytosomes and chlorogenic acid encapsulation, however, no reports were available about liposomes or phytosomes for yerba mate extract (not the isolated compounds). Also, references were updated as suggested.
Moreover, the methodology section is well organized and detailed. The reverse phase evaporation technique is correctly applied, and the characterization steps (UV-Vis, FTIR, DLS, zeta potential, and encapsulation efficiency) are standard and appropriate.
Re: We appreciate your comments about our methodology. We did some improvement considering reviewer #1 comments. All changes are red highlighted in the text.
Some improvements are suggested:
- In the preparation of extracts, it would be helpful to specify the standardization or characterization of the extract (e.g., total phenolic content or HPLC profile), to better relate it to its encapsulation performance.
Re: We agree with the reviewer suggestions. Unfortunately, we did not have an chromatography available at the moment of the experiment to report an in deep chemical composition of the extracts, our equipment is not working. However, we use a previous stablished method from our research group to produce the extracts (Dos Santos et al., 2023), using less leaves weight to have a diluted extract. The focus was to study and provide an encapsulation method. We did not determine how much extract (encapsulation efficiency) was encapsulated due to this limitation. We believe total phenolic compounds will be not meaningful in this context. In our future research we are exploring in deep extract stability and realising from phytosomes, and chemical composition will definitely be included and widely studied.
- When describing DLS and zeta potential analysis, please include how many replicates were performed and whether statistical differences were evaluated.
Re: Thank you for your comment regarding the replicates and statistical analysis in our DLS and zeta potential measurements. As noted in the Methods section, all analyses were performed following the manufacturer’s recommendations guidelines for nanomaterial characterization. The Zeta Sizer Nano Series (Malvern Instruments, UK) automatically conducts each measurement as an average of 12 consecutive scans to ensure reproducibility. For additional results reliability, we performed three independent replicates per sample (triplicate), with the final results reported as the mean ± standard deviation (SD) of these replicates.
Given the nature of our data in Table 1—where the measurements represent intrinsic physicochemical properties (e.g., size, PDI, zeta potential) rather than comparative experimental treatments—standard statistical tests (e.g., ANOVA or t-tests) were not applicable. This approach aligns with similar studies characterizing liposomes or nanoparticles, where triplicate measurements with mean ± SD are widely accepted to demonstrate trends and reproducibility (see references below). We appreciate the opportunity to clarify this point.
- https://doi.org/10.3390/nano11010171
- https://doi.org/10.1007/s00216-017-0527-z
- https://doi.org/10.3390/molecules25235655
- https://doi.org/10.1016/j.heliyon.2019.e02372
- Clarify whether any controls or blank liposomes were tested in parallel and how they were used in interpreting results.
Re: Liposomes without yerba mate extract were our controls. All results from these pure or purified PC liposomes are present in figures and tables. When was possible, also free yerba mate extract was analysed and the results were shown in figures (Fig. 2 ,3, 4 and 5).
Also, the results are clearly presented and illustrated with appropriate figures. The size distribution, encapsulation efficiency, and zeta potential data are coherent and support the formulation's stability and encapsulation success.
Re: Thank you for your comments and suggestions. We did the modifications as follow.
However, some points in the discussion require clarification:
- The authors observe that the presence of yerba mate extract results in smaller vesicle size and increased negative surface charge, and they hypothesize an interaction between phenolics and the phospholipid headgroups that may inhibit multilamellar structure formation.
- While this is an interesting and plausible idea, there is no experimental data in the manuscript (e.g., microscopy, molecular modeling, or comparative encapsulation using purified compounds) to support this conclusion.
- Therefore, this hypothesis should be clearly stated as a theoretical interpretation or further justified with literature support or a sentence indicating the need for future confirmation.
Re: Thank you for your in deep analysis of our results. Unfortunately, we do not have results from microscopy or molecular modeling. The comparation of encapsulation were performed using free-yerba mate extracts, non-loaded liposome and as references against loaded liposomes throughout UV-vis and FTIR. These techniques provided clear evidence of successful encapsulation, which was further supported by dynamic light scattering (DLS) and zeta potential (PZ) measurements. The significant changes in hydrodynamic diameter (DLS) and surface charge (PZ) between loaded and non-loaded liposomes confirmed the incorporation of yerba mate compounds into the liposomal structure.
Furthermore, the particle size associated with reverse phase method for liposome producing lead us to hypothesize the phytossome formation. We included some other literature references to support our claim. Also, we included the information “Although our data suggested this structural organization, further studies involving spectroscopic or microscopic techniques are necessary to confirm the formation of phytosomes. Moreover, phytosomes not only facilitate the controlled delivery of bioactive molecules but also enhance the solubility and bioavailability of these compounds, which can be beneficial in several applications, especially in beverages and food products, cosmetics, drugs and for agriculture uses”.
- The term “phytosome” is introduced in the conclusion but not previously explained or defined in the introduction or methodology. If the authors suggest that this system behaves similarly to phytosomes due to interactions between phenolics and lipids, this should be clarified earlier in the manuscript and be conceptually separated from standard liposomes.
Re: We included some new information about phytosome concept and some research about chlorogenic acids encapsulation trough this lipid vesicles. Please see lines 72-83.
- Consider discussing potential applications or benefits of these findings in real-world systems (e.g., food, nutraceutical, or cosmetic delivery) to increase the impact of the work.
Re: We included some examples of application of phytosomes throughout the results and discussion section (lines 237-240; 342-348) and at the conclusion (lines 365- 366)
Furthermore, the conclusion summarizes the findings well but as mentioned above, includes a speculative mechanism (lines 364–367) that is not directly supported by the results. A clearer statement like “Based on our observations, we propose that such interactions may occur, although further studies are required to confirm this mechanism” would be more appropriate.
Re: we included the sentence as suggested.
Comments on the Quality of English Language
The manuscript is understandable overall but would benefit from light editing for fluency and grammar. Some sentences are overly long or awkwardly structured. I recommend reviewing the manuscript with a native English speaker/colleague.
Re: The grammar and fluency were double checked and improved as suggested by a Canadian native speaker and retired English teacher.
Reviewer 3 Report
Comments and Suggestions for Authors
The article “A systematic preparation of liposomes loaded with yerba mate extract ”uses the reverse phase evaporation method to prepare liposomes loaded with yerba mate extract and conducts characterization studies on them, providing new ideas and methods for the protection and delivery of yerba mate's active components,but there are also some areas that need improvement. The specific review comments are as follows:
- The abstract is rather brief and lacks specific descriptions of the research results, merely touching upon vague concepts such as successful dispersion and membrane reconstitution.
- It is suggested to add comparative studies with other liposome encapsulation systems or direct application of yerba mate extract to highlight the advantages of this liposome.
- I wonder under what conditions the liposomes are most stable?
- Some of the references are relatively old.
- Standardize the reference format.
Author Response
Reviewer # 3
The article “A systematic preparation of liposomes loaded with yerba mate extract ”uses the reverse phase evaporation method to prepare liposomes loaded with yerba mate extract and conducts characterization studies on them, providing new ideas and methods for the protection and delivery of yerba mate's active components, but there are also some areas that need improvement. The specific review comments are as follows:
Re: We appreciate your time and comments. We reviewed carefully the manuscript and include as much detail as possible, as well new information as suggested. Please see all changes highlighted in red throughout the manuscript.
- The abstract is rather brief and lacks specific descriptions of the research results, merely touching upon vague concepts such as successful dispersion and membrane reconstitution.
Re: The abstract was rewritten to improve the information and results
- It is suggested to add comparative studies with other liposome encapsulation systems or direct application of yerba mate extract to highlight the advantages of this liposome.
Re: Thank you for your suggestions. We included some other studies from chlorogenic acid, catechin, epicatechin liposomes encapsulation to support the advantages of this liposomes. No yerba mate extract liposomes were produced up to now, according to our best knowledge.
- I wonder under what conditions the liposomes are most stable?
Re: Liposomes are very stable at refrigeration and pH around 7. However, our chromatography is not working, and we could not perform a stability study under different storage conditions. To have a proper study and access the individual compounds from the yerba mate extract and HPLC is needed. Unfortunately, we do not have these results to show at this moment and we included at conclusion some suggestions for future studies.
- Some of the references are relatively old.
Re: We updated the references from introduction and used them for discussion.
- Standardize the reference format.
Re: the reference format was double checked and standardized according to journals authors guideline.
Round 2
Reviewer 1 Report
Comments and Suggestions for Authors
The authors have improved the manuscript well and it is now ready for publication.